# Is Information Physical and Does It Have Mass?

**Mark Burgin [1] and Rao Mikkilineni [2,\*]**

1. Department of Mathematics, University of California, 405 Hilgard Ave. Los Angeles, Los Angeles, CA 90095, USA
2. Ageno School of Business, Golden Gate University, San Francisco, CA 94105, USA
* Correspondence: rmikkilineni@ggu.edu

**Abstract:** Some researchers suggest that information is a form of matter, calling it the fifth state of matter or the fifth element. Recent results from the general theory of information (GTI) contradict this. This paper aims to explain and prove that the claims of adherents of the physical nature of information are inaccurate due to the confusion between the definitions of information, the matter that represents information, and the matter that is a carrier of information. Our explanations and proofs are based on the GTI because it gives the most comprehensive definition of information, encompassing and clarifying many of the writings in the literature about information. GTI relates information, knowledge, matter, and energy, and unifies the theories of material and mental worlds using the world of structures. According to GTI, information is not physical by itself, although it can have physical and/or mental representations. Consequently, a bit of information does not have mass, but the physical structure that represents the bit indeed has mass. Moreover, the same bit can have multiple representations in the form of a physical substance (e.g., a symbol on a paper or a state of a flip-flop circuit, or an electrical voltage or current pulse.) Naturally, these different physical representations can have different masses, although the information is the same. Thus, our arguments are not against Landauer's principle or the empirical results of Vopson and other adherents of the physical nature of the information. These arguments are aimed at the clarification of the theoretical and empirical interpretations of these results. As the references in this paper show, recently many publications in which it is claimed that information is a physical essence appeared. That is why it is so important to elucidate the true nature of information and its relation to the physical world eliminating the existing misconceptions in information studies.

**Keywords:** information; physics; general theory of information; material carrier; material representation; knowledge; mass–energy–information–knowledge correspondence

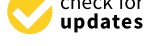



## 1. Introduction

Physical science is a branch of natural science that studies non-living systems, in contrast to life science, which studies living things. On the other hand, information science, according to the Merriam-Webster dictionary, is primarily concerned with the analysis, collection, classification, manipulation, storage, retrieval, movement, dissemination, and protection of information. However, while mathematicians, philosophers, biologists, physicists, and information scientists, to mention but a few, have all postulated various definitions of information since the notion of information emerged in human society, it is not an exaggeration to say that there is no consensus on what information really is.

Does information exist independently of our own existence? Does information processing require only living organisms, or also other material structures in the physical world to process information? Unlike humans, do the technical information-processing structures know that they are processing information? How is knowledge related to the information? While these are profound questions, the purpose of this paper is not to answer them. For answers, we refer the reader to the general theory of information (GTI) in [1–9] and in other related publications where these questions are studied and the answers are obtained. We use this theory in this paper because it is demonstrated that GTI gives the

most comprehensive definition of information, encompassing and clarifying what other researchers wrote about information.

In this paper, we investigate the mass-energy–information equivalence principle suggested in [10–14] and the related claims that information is physical, has mass, and is the fifth state of matter. "*For over 60 years, we have been trying unsuccessfully to detect, isolate or understand the mysterious dark matter," said Vopson. "If information indeed has mass,*" he continued, "*a digital informational universe would contain a lot of it, and perhaps this missing dark matter could be information*" [10]. This statement is based on the mass-energy–information equivalence principle, which was suggested by Vopson and claims that information is transformed into mass or energy depending on its physical state. In addition, the existence of the intrinsic information underpinning the fundamental characteristics of elementary particles in the universe implies that stable, non-zero rest mass elementary particles store fixed and quantifiable information about themselves [10–22]. These so-called information conjectures also seem to imply that the information is a form of matter, which is called the fifth state of matter or the fifth element by Vopson [11–14].

It is necessary to remark that while the suggestion that information has mass is not accepted by many researchers, many of them, and the majority of lay people, think that information is physical by its nature. Thus, the main goal of our paper is to explain that this is not true and elucidate the true nature of the information.

To validate the assertions about the physical nature of information together with similar claims, we analyze the assumptions behind the formulated mass–energy–information equivalence principle using the GTI, and demonstrate that information is not physical by itself but has a physical representation. Naturally, this physical representation has mass and complies with physical laws.

In contrast to this, Landauer wrote:

> "*Information is inevitably inscribed in a physical medium. It is not an abstract entity. It can be denoted by a hole in a punched card, by the orientation of a nuclear spin, or by the pulses transmitted by a neuron. The quaint notion that information has an existence independent of its physical manifestation is still seriously advocated [23]. This concept, very likely, has its roots in the fact that we were aware of mental information long before we realized that it, too, utilized real physical degrees of freedom.*" [17] p. 64

We argue that the physical properties that Landauer [15–18], Vopson [11–14], and other researchers ascribe to information [19–22] are actually the properties of the physical representations of information.

Note that while other researchers also repudiated the physical nature of information, nobody described the correct place of information in the world (cf., for example, [23–25]), while the general theory of information explains where information, in the strict sense, exists. It is important to understand the difference between information and its physical carrier because different physical carriers can contain the same information. Various observations support this statement.

Information per se belongs to the world of structures and does not have mass, but its representation (carrier) in the form of a physical structure possesses mass. In the physical world, genes, and neurons, for example, process information to convert it into knowledge. They communicate information, which is represented as biological and neurological structures, using chemical or electrical signals. In the digital world, a 'bit' of information does not exhibit mass, but a physical material that represents the bit indeed has mass. The same bit can have multiple representations in the form of physical material (e.g., a symbol on a paper or a state of a flip-flop circuit, or an electrical voltage or current pulse). Information is carried by the physical structures in the same way thermometers "carry" temperature.

Thus, the physical properties that Landauer and other researchers deduced, ascribing them to information [10–22], are actually the properties of the physical representation of information. This is in good agreement with what Landauer actually wrote, stating that "*information is inevitably tied to a physical representation,*" and not with his more far-reaching claims such as "*information is a physical entity*" [17] p. 64.

It is necessary to remark that our arguments are not against Landauer's principle or the empirical results of Vopson. They are useful scientific results (cf., for example [26–28]). Our goal is the clarification of the theoretical and empirical interpretations of these results, explaining that they are about carriers of information but not about information itself. It is important to understand that the properties of information representations and information carriers are very important because people do not interact directly with information but work, for example, computing only with information representations and information carriers.

At the same time, it is important to know how to derive properties of information from properties of information representations and information carriers, and the general theory of information (GTI) provides efficient means for performing this. For instance, the whole area of cryptography studies how to find good information representations for secure transmission and preservation of information (cf., for example, [29,30]. Coding is a transformation of information representations and sometimes of information physical carriers. Programming is also a transformation of information representations. In this respect, our results complement the results of Landauer, Vopson, and other researchers who study the properties of physical information representations and information carriers.

The paper has the following structure. In Section 2, we present the ideas and conceptions from the GTI about information, its representation, and the relationship between information and knowledge. In Section 3, we discuss the mass-energy–information equivalence principle in light of the GTI. In Section 4, we put forward general observations from this study and conclusions.

## 2. General Theory of Information

The general theory of information (GTI) [1,3] states that "knowledge is related to information as the matter is related to energy". At the same time, the material structures in the physical world carry the information that represents the state and the dynamics of the structure under consideration. In the physical world, material structures are governed by the transformation laws of matter and energy. Energy has the potential to create or change material structures. All physical and chemical structures, which are created or changed by the transformation of matter and energy, obey the laws governing their transformations. All physical structures contain potential information that characterizes their structure, the functions of their constituent parts interacting with each other and with their surroundings, and their behaviors when internal and external factors cause fluctuations in their interactions. In fact, it means that there is a definite relationship between the characteristics of physical objects allowing the possibility of the conversion of mass into the energy of physical objects described by these characteristics. The famous formula $E = mc^2$ connects the energy and mass of physical objects. However, in contrast to what many people think, this formula does not mean that substance (matter) is equal to energy, but it shows the maximal amount of energy in a physical object with a given mass.

The states of physical structures and the regularities of their evolution are described by the laws of physics, which are mental structures created by humans (mainly by physicists and mathematicians). Living organisms have developed physical structures that exploit matter and energy transformations to acquire a unique identity and the ability to sense and process information that is carried by material structures and convert it into knowledge in the form of mental structures. While all living organisms have varying degrees of the ability to perceive, process, and convert information into knowledge, humans have developed the highest level of representing and managing mental structures using ideal structures in the form of named sets or fundamental triads [1]. The fundamental triad provides the schema and operations to create knowledge in the form of entities, their relationships, and their evolution consisting of event-driven behaviors [7–9]. Events are caused by fluctuations in the interactions among the components of the structures and their interaction with their environment. Thus, functions, structure, and fluctuations play important roles in the system's microscopic and macroscopic behaviors [31].

It is important to note that the mental models created by processing information are observer-dependent, as they depend on the previous knowledge of the observer in addition to many other idiosyncratic factors.

According to [1,3], the GTI places information per se in the ideal world of structures, which is the scientific manifestation of the world of Plato's Ideas or Forms [4]. Namely, the concept *structure* provides the scientific representation of Platonic Ideas, while the existence of the world of structures, which can be naturally equated to the world of Plato's Ideas, is proved by scientific means.

According to the ontological principle, O2 and its additional forms in the GTI ([1] (p. 99), [3]), information plays the same role in the world of structures as energy plays in the physical (material) world. While being associated with material structures in the physical world, the information does not belong to this world and can only be materialized in a physical form as asserted in the GTI [2]. Relations between information and structures were also considered by Stonier, who claimed that information has the power to exhibit itself as a structure when added to matter [32,33].

According to the ontological representability principle (ontological principle O4) of the GTI ([1] (p. 123), [3]), for any portion of the information I, there is always a representation Q of this portion of information for a system R. Often this representation is material, and as a result, since information is materially represented, many people comprehend information as physical. Consequently, a physical representation of information can be treated as the materialization of this information [2]. Thus, information not being physical by itself has a physical representation, and naturally, this physical representation complies with physical laws.

Moreover, according to the ontological embodiment principle (ontological principle O3) of the GTI ([1] (p. 120), [3]), for any portion I of information, there is always a carrier C of this portion of information for a system R. This carrier is, as a rule, material, and this makes information even more present in the physical world. The physical carrier of information can also be treated as the materialization of this information, or more precisely, the materialization on the second level. Materialization of information can require an agent or an observer to perform the process of materialization. An example is representing information in the form of symbols on the carrier, which is a piece of paper using a pen as a tool for materialization.

To show the difference between carriers and representations of information, we explain that any physical representation of information is also a physical carrier of the same information. A physical carrier of a portion I of information is any physical thing that contains this portion of information. At the same time, a physical representation of a portion I of information is such a physical carrier that allows direct extraction of this information. Thus, any physical representation is a physical carrier, but not any physical carrier is a physical representation. For instance, an envelope is the physical carrier of information contained in the letter this envelope encloses, the piece of paper on which the text of the letter is printed or written is also a physical carrier of the same information, and, finally, the text of the letter is also a physical carrier of the same information. However, direct extraction of information is possible only from the text. We cannot extract this information from the envelope or the piece of paper without the text. Consequently, the envelope that contains this letter or the paper on which this text is printed or written, as well as this piece of paper, are only carriers but not representations of the information in this letter.

The carrier of the information I that is not a representation of this information is called the enveloping carrier of I.

In the mental world created by living organisms, information received from the environment using the five senses enables mental representation and is converted into mental structures formed of fundamental triads [1]. There are two forms of mental structures —those that are derived from external observations and those that are created by the human mind representing the ideal structures. Mathematics is used to represent the ideal structures

and operations with them, as well as to model the systems from the material world, their states, and evolution.

Similarly, the mental reality (mental world) consists of various mental structures, which participate in the transformational processes involving information and knowledge. These transformational processes are defined by the physical information-processing structures, which consist of genes and neurons. The formula that is similar to Einstein's mass–energy equivalence also exists in the information realm of mentality.

To elaborate on this formula, it is necessary to explain that knowledge in the strict sense belongs to the world of structures, because knowledge consists of knowledge structures. At the same time, similar to information, knowledge has representations and carriers in the material (physical) and mental worlds. Various books and journals are physical carriers of knowledge, containing different knowledge representations. For instance, a formula, such as $E = Mc^2$, in the textbook in physics is a mathematical representation of knowledge about physical reality. Mental representations of knowledge exist in the mentality of the people and the mentality of groups of people, such as the community of physicists or mathematicians. However, many people call by the name knowledge what is really the mental representation of knowledge.

With this in mind, we introduce a new characteristic of mental knowledge named mental knowledge mass. Namely, the mental mass MK of a mental knowledge unit K is the measure of the knowledge object inertia concerning the structural movement in the mental world. Each mental knowledge mass reflects properties of the structural components of mental knowledge, their relationships, and behaviors. One mental knowledge structure interacts with other mental knowledge structures by sharing information using various means of communication facilitated by the information-processing physical systems such as genes and neurons, which use chemical and neuronal signals for communication.

Based on the concept of mental knowledge mass, we obtain the equivalence formula, which has the form I = MK*p, where $p > 0$ is the constant that connects the information I and mental knowledge K of mental systems just as energy and matter are connected in the physical world. This is a theoretical conjecture, which needs experimental validation. Finding the numerical value of the constant p could allow the estimation and measurement of information contained in mental knowledge systems.

With respect to mental mass, it is important to understand that mental knowledge has mental mass but not knowledge and information, which belong to the world of structures. Besides, energy, which is the physical counterpart of information, also does not have mass but only its measure is proportional to the mass of physical objects.

As the result, we arrive at the equivalence between the theory of physical structures and the theory of mental structures. Each such structure with a certain mass interacts with other structures based on various relationships defined by interaction potentials. In such a way, each structure provides guidelines for functional behavior and a network of structures provides guidelines for collective behavior based on interactions between structures. Wired together structural nodes of the network also fire together, shaping the collective behavior of the system. This allows us to represent the mental structures using the same mathematical representations of physical structures in the form of state vectors and their evolution.

In this context, a knowledge network is an assembly of components with specific functions, which interact as ideal structures and produce a stable behavior (equilibrium) when conditions are right. However, fluctuations change the interactions and cause non-equilibrium conditions. This leads to emergent behaviors leading to chaos. However, biological systems have developed an overlay of information-processing structures, which support and manage the system stability, safety, sustenance, etc., while monitoring the impact of environmental fluctuations.

### 3. GTI and the Mass–Energy–Information Equivalence Principle

Armed with this knowledge about information, we can now respond to the questions: is information physical and does it have mass? Answering the first question, we explain that information is associated with physical and mental structures, as its representations and carriers are embedded in other physical and mental structures that act as carriers of information. Answering the second question, we conjecture that the knowledge in mentality has a mental mass just as the matter has physical mass, while the information carriers (both physical and mental) have physical or mental mass but not the information itself.

These conclusions put us at odds with those researchers who claim information has mass [10–19]. For instance, Landauer claims that information is physical. However, at the beginning of his paper [17] p. 64, he writes

*"Information is inevitably tied to a physical representation."*

It means that, according to Landauer, information is only tied to its physical representation but this tells nothing about the essence of information per se.

Another statement from his work is:

*"Information is not a disembodied abstract entity; it is always tied to a physical representation" asserts what information is not telling anything of what information per se is".* [16] p. 188

Similarly, Melvin Vopson claims

*"A computational process creates digital information via some sort of physical process, which obeys physical laws, including thermodynamics."* [11]

As we explained before, this statement is misleading. The correct statement should be:

*"A computational process creates digital information via some sort of physical process, which works with physical representations of digital information and obeys physical laws, including thermodynamics."*

Namely, only by changing physical representations, the physical process changes information [2]. In particular, erasing information changes the physical objects that were carriers of this information, while writing information transforms some physical objects into the carriers of the written information.

For instance, the Landauer principle states that logically irreversible computation can be only implemented by thermodynamically irreversible processes. In this setting, logical or abstract computation is performed with linguistic (symbolic) representations of information, while physical computation operates with physical representations and carriers of information [26].

Accordingly, the Formula (6) from [11] can be interpreted not as the mass of a bit of information, but as the mass of the physical representation of a bit of information.

Besides, there is a problem with the interpretation of Shannon's measure of information (information entropy) H. It measures information not directly but utilizes information's physical representations—signals or texts. When this measure is applied to the states of physical systems, it means that the state of a physical system is a representation of information while the corresponding system is the carrier of this information.

As the result, the mass-energy–information equivalence conjectured by Vopson in [11] is not valid because the same portion of information can have different physical representations. In other words, the mass and energy of the different representations of the same information can vary.

This situation is clearly explained by the general theory of information (GTI) mentioned above. Indeed, according to the ontological principle O4, for any portion of information I, there is always a representation Q of this portion of information for a system R [1,3]. Often this representation is material, and as a result, being materially represented, information becomes, in some sense, physical. In this context, a physical representation of information becomes the materialization of this information allowing people and other

systems to obtain this information [2]. For instance, the process of DNA replication shows that not only living beings but also unanimated systems such as molecules can transform and transmit information from one physical representation to another one.

Thus, information is not physical by itself but has a physical representation and, naturally, this physical representation complies with physical laws. This is in good agreement with what Landauer actually wrote in some of his works and not with his and his adherents' more far-reaching claims.

Similarly, some people can say that thoughts or feelings are physical because they are in the brain, which is physical. However, according to contemporary psychology, the brain is only the carrier of thoughts and feelings, the nature of which is essentially not physical. In particular, thinking is defined as "a mental process that involves the manipulation of information" [34].

One more argument that demonstrates that information is not physical is presented in [23]. Based on the conjecture of the physical nature of information, Kosso asserts that "information is transferred between states through interaction" because physical influences can be transferred only through interactions [35]. However, this assertion, for example, contradicts the so-called 'negative experiments' [36] where an object or event can be observed by noticing the absence of another object or event. It means that the observer obtains information without interaction with the object or event that contains this information [37].

Thus, the physical properties that Landauer, Vopson, and other researchers ascribe to information [10–19] are actually the properties of the physical representations of information.

## 4. Where Information Belongs

There are also other researchers who explain that information is not physical. For instance, Timpson justifies that the claim "information is physical" is essentially wrong because the term information "doesn't serve to refer to a material thing or substance" [38–40]. To make his approach complete, Timpson suggests that information exists in the form of "pieces of information, quantum or classical," while these pieces of information "are abstract types" and "they are not physical" [38]. This understanding is also supported in [23].

To understand the pitfall of this approach, we need to know what an abstract object or abstract type is. Philosophers elaborated a theory of abstract objects and abstract type (cf., for example, [41–43]). The main underpinning of this theory is the distinction between abstract and concrete, which did not play a noteworthy role in philosophy before the 20th century. However, in the 20th century, abstraction came to the forefront of mathematics and science. As a result, several philosophers tried to elaborate a clear and exact form of the notion of *abstract objects*, but mostly concluded that ordinary objects, such as trees and tables, are possibly concrete, while abstract objects, such as number 1 or straight lines, are not concrete [43].

Although the modern distinction between abstract and concrete objects bears some resemblance to Plato's differentiation of Ideas and Sensibles, this only conflates the concepts of ideal and abstract without any well-grounded reason. Being unable to find an explanation of the ideal reality of Plato, some philosophers decided to change the term *ideal* to the term *abstract* as contemporary science and mathematics went to higher and higher levels of abstraction, making the latter term more comprehensible.

In this context, the most reasonable approach to abstract objects is assuming that an abstract object consists of a name and a set of properties [44]. Based on this assumption, Edward Zalta built a formal axiomatic theory of abstract objects [43]. However, this theory does not answer the question about the place of abstract objects in the world.

Let us try to answer this important question. It is natural to suppose that as their name suggests, abstract objects are formed in the process of abstraction. This is an elaborate mental process that goes through various stages and achieves different levels of abstraction [45]. This situation implies that abstract objects as results of abstraction dwell in mentality. Some

of them belong only to individual mentality, while others also come to the group and social mentalities.

This correlates with the opinions of philosophers. For instance, Falguera, Martínez-Vidal, and Rosen write:

> "*The modern distinction* [between abstract and concrete, M.B. & R.M] *bears some resemblance to Plato's distinction between Forms and Sensibles. But Plato's Forms were supposed to be causes par excellence, whereas abstract objects are generally supposed to be causally inert.*" [41]

Now we can analyze and answer the question of whether the information is an abstract object. We know definitely that there is information in mentality but the GTI also tells us that there is ontological information, which exists in nature being wrath encapsulated in physical systems independently of any mentality [46–48]. This persuasively shows that information is not an abstract object but, as the GTI demonstrates, it belongs to the world of ideal structures and comes to the physical and mental worlds through materialization and mentalization [2]. In particular, abstract objects are mental representations of information from the world of ideal structures.

Discussing abstract objects, it is important to understand that being names, with properties, which can be described by axioms in the formalized setting, abstract objects are special kinds of structures, and namely, they are external structures in the sense of the general theory of structures [5].

## 5. Conclusions

As it is possible to see from the discussion above, information is not physical by itself but has a physical representation and, naturally, this physical representation complies with physical laws. This is in good agreement with what Landauer actually wrote and not with his more far-reaching claims. Thus, the physical properties that Landauer and other researchers conjectured, ascribing them to information [10–19], are actually the properties of the physical representation of information.

The argument of Vopson that "*Archibald Wheeler . . . postulated that the universe emanates from the information inherent within it and he coined the phrase "It from bit"* [12] does not prove the physical nature of information because, according to the GTI, coming from the world of structures, information has a strong impact on the physical world [48].

Recently there were many publications in which it is claimed that information is a physical essence [10–19,49–53]. That is why it is so important to elucidate the true nature of information and its relation to the physical world eliminating the existing misconceptions in information studies.

In addition to this paper, the true nature of information and its relation to physical reality is also explained in [1–5] and related publications. It is possible to explain this only based on the GTI because there is no other theory of information in which it is proven that information, in the strict sense, belongs to the world of ideal structures.

It is important to emphasize the conclusions drawn from GTI. Information plays an important role in describing the material structures in the physical world, as well as the mental structures in the mental world created by biological systems through evolution and natural selection. Both material structures and mental structures are involved in receiving information, processing information, and communicating information. Information, in essence, describes the state of a structure and its evolution when events change it. The state of a material structure and its evolution are governed by the transformation laws of energy and matter. The information about a material structure can also be materialized and communicated using information carriers, which are also material structures. Communication of information using material structures, therefore, also obeys the transformation laws of matter and energy. While information per se has no mass, the materialized information (e.g., a symbol on a paper or a state of a flip-flop circuit, or an electrical voltage or current pulse) has mass. On the other hand, information received by the biological systems is processed and converted into knowledge in the form of mental structures. These mental

structures are materialized in the form of multi-layered networks of genes and neurons. Genes use sequences of symbols (DNA) and neurons use sub-symbolic computing to process information and use the knowledge to execute "life processes." These mental processes are distinguished by the self-organizing properties of biological systems in contrast to the dynamics of material structures subject to the laws of physics or behave the same way as complex adaptive systems that can exhibit emergence under fluctuations. GTI provides the tools to model both the material and mental structures and describe the conversion processes transforming information and knowledge.

To conclude our discussion, we remind the reader that mathematicians were able to understand the difference between numbers and their representations by numerals a long time ago. Hopefully, information scientists and other researchers will also be able to understand the difference between information and its physical representations. More importantly, they will be able to use the GTI to improve how we use information and knowledge, as well as to enhance our understanding of how nature operates and additionally design the digital world, which would imitate living organisms with such behaviors as autopoiesis and cognitive reasoning [7–9].

**Author Contributions:** Conceptualization, M.B. and R.M.; Formal analysis, M.B.; Investigation, R.M.; Writing—original draft, R.M.; Writing—review & editing, M.B. and R.M. All authors have read and agreed to the published version of the manuscript.

**Funding:** This research received no external funding.

**Institutional Review Board Statement:** Not Applicable.

**Informed Consent Statement:** Not Applicable.

**Data Availability Statement:** Not Applicable.

**Conflicts of Interest:** The authors declare no conflict of interest.

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
