# Peer review of "Is Information Physical and Does It Have Mass?"

_information, doi:10.3390/info13110540_

Round 1

Reviewer 1 Report

The work deals with a fascinating yet complex subject. There are some things in the text that I don't understand and that need to be explained, in my opinion.

In chapter 3 it is denied that information has mass, for quite very convincing reasons. But if the information has no (physical or mental mass) how can knowledge have? Can you specify better what is mental mass? What is the ontological difference between knowledge and information when they both appear to be immaterial?

The criticisms to the advocates of the physicality of knowledge must be more detailed; for example, the critique of Landauer [line 240] cannot be entrusted to a single sentence. Similarly, quoting formula (6) [line 263] is not much help. The author should say something more so that the reader is not forced to read the work [11]. The same applies to the reference to Figure 2 [line 271].

Section 4 is not conclusive in my opinion while the author is quite clear regarding the negative characterizations of  information he is rather laconic for the positive ones [lines 338-342]

Author Response

Thanks to the reviewer for the suggestions. We have expanded the explanation of mental mass and added quotes from various sources to enhance the context of the criticism. Also addressed suggestions on section 4.

Reviewer 2 Report

This reviewer needs little convincing that info is not material, but rather is reflected in material structures, such as the carrier and the representation.  She suggests that the analysis via GTI sometimes obscures that clear statement, and that the tools of GTI be used judiciously as support rather than superstructure.  In other words, the authors should directly address the question whether their thesis is justifiable even without GTI but based on the same principles that led to GTI.

And while they state plainly that the targets of their criticism are interpretations and external claims about Landauer's and Vopson's work, no citations are given for those claims.  The quotes from line 237 on can be aligned with the authors' view of info.  So what exactly are the claims that info has mass (line 237)?

Line 77:  Information is carried by the physical structures in the same way as bottles carry water.

>>>  Beware analogy with water, which has mass.  In the same way that thermometers reflect/carry temperature?

Line 108:  Delete "is", delete "the".  Replace "while" with ";"?

Line 125:  I think the authors mean that the laws of physics ARE mental structures.

Line 132: Delete "the"

Line 140:  ... in addition to many other idiosyncratic factors.

Line 150:  Delete "is"

Line 157-8:  This claim undercuts the thesis even qualified by "in some sense"

Line 173:  "but not any carrier of some information I is" would be more clear as "but not all carriers of some information I are"

Line 179-80:  Remove commas.

Line 182:  "obtains" is too active; perhaps "affords" or "allows" or "enables"

The paragraph starting line 189 is intriguing, but seems to align info very closely to mass, via knowledge.  Is a knowledge unit K actually physical, subject to inertia?  If M_k (line 204) is mass, that suggests that I is mass.  The notion of "mental mass" (line 234) needs clarification to distinguish it from physical mass even if it is well-described in the cited works.

Lines 254-257 make the authors' point clearly.  Good.

Lines 289-90: This is an assertion, not an argument, and should not appear in scholarship unless substantiated.

Lines 299-300:  "and other researchers deduced 299 ascribing them to info" should be "and other researchers ascribe to info"

Line 335:  Remove "the", as it suggests that you refer to some particular info that the reader should know about.

Section 4:  The discussion on the issue of whether info is abstract is quite interesting, and could use more development and exposition.

Line 345:  Delete "and"

Line 378:  The verb "remind" requires and object, such as "we remind the reader" 

Author Response

We are grateful to the unknown reviewers for the useful remarks and suggestions. Taking them into account, we made the following changes in our manuscript “Is Information Physical and does it Have Mass?”

  1. New citations from Landauer's and Vopson's works are added to give a better explanation of their views.
  2. Explanations were added at the places that had been unclear.
  3. The style was improved based on the suggestions of the reviewers.
  4. Misprints were corrected.

Sincerely,

Dr. Mark Burgin and Dr. Rao Mikkilineni

Round 2

Reviewer 2 Report

In my field of computer science, no one would dream of claiming that information has mass, so the refutation is not obviously salient.  Without the comfort of familiarity with the niche question here, I have difficulty distinguishing between argument, based on clearly stated premises, and assertions, which appear to be common knowledge shared with the expected reader.

The assertions start with the GTI itself, and how much of the thesis of this paper relies on it.  Discussion on that very point would still be useful.  And I find no definition of "carrier" that would explain why some carriers are the same as the representation and some are not.  Etc.

Author Response

Taking into account the remarks of the reviewer, we added explanations at the places that were not sufficiently clear according to the reviewer. We have also checked for English language and grammar using the tool Grammarly.

  1. The reason for this paper is explained
  2.  More explanation is added to clarify the points made by the reviewer

Sincerely,

Dr. Mark Burgin and Dr. Rao Mikkilineni